# Liquid Biopsy Analysis as a Tool for TKI-Based Treatment in Non-Small Cell Lung Cancer

**DOI:** 10.3390/cells11182871

**Published:** 2022-09-14

**Authors:** Karolina Buszka, Aliki Ntzifa, Barbara Owecka, Paula Kamińska, Agata Kolecka-Bednarczyk, Maciej Zabel, Michał Nowicki, Evi Lianidou, Joanna Budna-Tukan

**Affiliations:** 1Department of Histology and Embryology, Poznan University of Medical Sciences, 60-781 Poznan, Poland; 2Doctoral School, Poznan University of Medical Sciences, 60-812 Poznan, Poland; 3Analysis of Circulating Tumor Cells Lab, Lab of Analytical Chemistry, Department of Chemistry, National and Kapodistrian University of Athens, 15771 Athens, Greece; 4Department of Immunology, Chair of Pathomorphology and Clinical Immunology, Poznan University of Medical Sciences, 60-806 Poznan, Poland; 5Division of Anatomy and Histology, University of Zielona Góra, 65-046 Zielona Góra, Poland

**Keywords:** circulating tumor cells (CTCs), cell-free DNA (cfDNA), exosomes, tumor-educated platelets (TEPs), liquid biopsy, non-small cell lung cancer (NSCLC), tyrosine kinase inhibitors (TKIs)

## Abstract

The treatment of non-small cell lung cancer (NSCLC) has recently evolved with the introduction of targeted therapy based on the use of tyrosine kinase inhibitors (TKIs) in patients with certain gene alterations, including *EGFR*, *ALK*, *ROS1*, *BRAF*, and *MET* genes. Molecular targeted therapy based on TKIs has improved clinical outcomes in a large number of NSCLC patients with advanced disease, enabling significantly longer progression-free survival (PFS). Liquid biopsy is an increasingly popular diagnostic tool for treating TKI-based NSCLC. The studies presented in this article show that detection and analysis based on liquid biopsy elements such as circulating tumor cells (CTCs), cell-free DNA (cfDNA), exosomes, and/or tumor-educated platelets (TEPs) can contribute to the appropriate selection and monitoring of targeted therapy in NSCLC patients as complementary to invasive tissue biopsy. The detection of these elements, combined with their molecular analysis (using, e.g., digital PCR (dPCR), next generation sequencing (NGS), shallow whole genome sequencing (sWGS)), enables the detection of mutations, which are required for the TKI treatment. Despite such promising results obtained by many research teams, it is still necessary to carry out prospective studies on a larger group of patients in order to validate these methods before their application in clinical practice.

## 1. Introduction

Lung cancer is the leading cause of cancer-related death in the world [1,2,3]. In 2020, it was responsible for 1.8 million deaths (18.0% of the total cancer deaths) preceding colorectal (9.4%), liver (8.3%), stomach (7.7%), and female breast (6.9%) cancers. Moreover, lung cancer ranks second for incidence in both sexes worldwide with an estimated 2.2 million new cancer cases in 2020. In men, lung cancer is the most frequently diagnosed cancer and the main cause of cancer death. In women, however, it comes second for mortality and third for morbidity. It is reported that incidence and mortality rates are approximately two times higher in men than in women [3].

There are over 50 histomorphological subtypes of lung cancer, among which, non-small cell lung carcinoma (NSCLC) and small cell lung carcinoma (SCLC) are the two main types. NSCLC occurs in roughly 80–85% of patients with lung cancer, whereas SCLC comprises 15% of cases [4]. According to WHO classification, NSCLC includes a variety of different subsets with three most notable ones: adenocarcinoma, squamous cell carcinoma, and large cell carcinoma [1,2,4,5,6]. Adenocarcinoma is the predominant subtype of NSCLC and represents approximately 40–50% of lung cancers. Squamous cell carcinoma accounts for 25–30% of cases, whereas large cell cancer is responsible for approximately 5% to 10% of all types [1,4].

The prognosis for lung cancer patients is poor; however, it strongly depends on the stage of the disease at the moment of diagnosis. In particular, the 60-month overall survival (OS) rate for NSCLC ranges from 68% in patients with stage IB disease to 0% to 10% in patients with stage IVA-IVB disease [1]. For this reason, new technologies for the early and precise detection of lung cancer are urgently needed to enable the prompt introduction of targeted therapy, with particular emphasis on personalized treatment.

## 2. Therapeutic Approaches in NSCLC

The treatment of NSCLC depends on the stage of the disease. Patients with stage I or II should be treated with complete surgical resection, if possible [1]. Unfortunately, surgical resection is not feasible in over 60% of patients that present locally advanced or metastatic disease (stage III or IV) at the time of diagnosis. Until recently, conventional chemotherapy and radiation therapy were the main ways of treatment for these patients [4]. Although adjuvant platinum-based chemotherapy is a standard treatment for stages II-IIIA disease, relapse rates are high, with a relatively high rate of toxicity [7,8]. Moreover, platinum-based chemotherapy has provided only a modest survival benefit for advanced NSCLC patients, with an OS of less than 2 years [9].

The treatment of lung cancer has recently evolved with the introduction of targeted therapy based on the use of tyrosine kinase inhibitors (TKIs). TKIs are used in patients with metastatic NSCLC (stage III or IV) and certain gene alterations, including *EGFR, ALK, ROS1, NTRK1, BRAF, HER2, MET*, and KRAS genes, which are independent of PD-L1 levels. Patients with metastatic NSCLC, targetable driver oncogene molecular variant, and also PD-L1 expression levels of 1% or more should receive first-line targeted therapy for detected oncogene instead of first-line immune checkpoint inhibitors (ICIs). Targeted therapies yield higher response rates (e.g., osimertinib, 80%) than ICIs (poor response rates) in the first-line setting and are better tolerated [10,11,12,13]. Molecular targeted therapy based on TKIs has improved clinical outcomes in a large number of NSCLC patients with advanced disease, enabling significantly longer progression-free survival (PFS) [1,8,9,14]. TKI treatment side effects are fewer and less serious as compared with conventional chemotherapy and they include: skin lesions, diarrhea, general malaise, and hepatotoxicity, and are usually reversible. The only potentially lethal side effect of TKI treatment is interstitial lung disease; however, it occurs very rarely [15]. Among mentioned genes used as targets in the TKI therapy of NSCLC, *EGFR*, *ALK*, *ROS1*, *BRAF*, *MET*, and *KRAS* should be noted [16]. A summary of the TKIs used depending on the detected mutation is shown in Figure 1.

### 2.1. EGFR-Targeted TKI Treatment

#### 2.1.1. Major EGFR Mutations

Activating mutations in the tyrosine kinase domain of the epidermal growth factor receptor (*EGFR)* gene occur in approximately 10–30% of NSCLC tumors [17]. They affect most commonly exons 18–21 of the *EGFR* gene, causing the ligand-independent activation of the tyrosine kinase of *EGFR* [18] and uncontrolled cell proliferation which may result in malignant transformation.

It is reported that the majority of patients with *EGFR*-mutated NSCLC (80% to 90%) have either an exon 19 deletion or an L858R point mutation [19]. The deletions in exon 19 include leucine (L) at codon 747, arginine (R) at codon 748, glutamic acid (E) at codon 749, and alanine (A) at codon 750 (∆LREA). The point mutation in exon 21 results in the substitution of leucine (L) to arginine (R) at codon 858 (L858R) [9,20].

In patients with activating *EGFR* mutations, the inhibition of the tyrosine kinase activity of *EGFR* may provide a significant anti-tumor effect. The most commonly used and approved by Food and Drug Administration (FDA) *EGFR* TKIs include: erlotinib (Tarceva), gefitinib (Iressa), afatinib (Gilotrif), and osimertinib (Tagrisso) [14]. In general, mutations occurring in exons 18 to 21 are responsive to *EGFR* TKIs; however, an *EGFR* T790M mutation in exon 20 is associated with acquired resistance to TKI therapy. This mutation involves a substitution of threonine to methionine in exon 20 and may affect up to 63% of patients with disease progression after the initial response to front-line TKIs. In such cases, it is recommended to introduce third-generation *EGFR* inhibitors (osimertinib and rociletinib) [14,19]. Interestingly, osimertinib was quite recently approved as a first-line treatment for patients with metastatic *EGFR-*mutant NSCLC previously untreated [21] and also as an adjuvant treatment for NSCLC with *EGFR* Ex19del or L858R mutations for stages IB−IIIA [22].

#### 2.1.2. Rare EGFR Mutations

Despite the fact that the most common *EGFR* mutations in NSCLC are L858R substitution and exon 19 deletion, 10% of NSCLC patients will have an uncommon *EGFR* mutation. Generally, mutations involving exons 18 to 21 are considered sensitive to *EGFR* TKIs, with the exception being mutations involving exon 20, including T790M and exon 20 insertions [19].

Mutations in exon 18 are typically considered sensitizing to *EGFR* TKI therapy. The most frequently detected exon 18 mutation is the G719X mutation [23,24]. Compared with wild-type *EGFR*,53, the G719X mutation is associated with a 10-fold increase in *EGFR* activation. However, in vitro studies have suggested that it is not as sensitive to gefitinib as NSCLC cell lines with L858R mutations [25]. Patients with G719X mutations that respond to *EGFR* TKI therapy have not always been as prolonged as those seen with the more common mutations.

However, complex mutations within exon 18 (e.g., L861Q mutation) may be associated with a better prognosis than point mutations [24,26]. Yang et al.’s study, in which 12% of patients had L861Q substitutions, has shown the association of the presence of the L861Q mutation with sensitivity to *EGFR* TKIs [18,27]. For those patients, the objective response rate (ORR) was 56.3% with a median progression-free survival (PFS) of 8.2 months and overall survival (OS) of 17.1 months [28].

Another mutation, the S768I mutation, in some studies [29,30], is associated with poor responses to TKIs. However, patients with the S768I mutation from treatment with afatinib did show good clinical outcomes, with a median PFS of 14.7 months and median OS not yet reported (95% CI: 3.4 months–not estimable) [28].

### 2.2. ALK-Targeted TKI Treatment

Anaplastic lymphoma kinase (*ALK*) gene rearrangements are present in approximately 5% of NSCLC tumors [31]. One of the most frequently described rearrangements is an inversion of the short arm of chromosome 2, which results in a fusion of *ALK* with the echinoderm microtubule-associated protein-like 4 (*EML4*) gene. The fusion protein EML4-ALK activates several pathways driving cell survival and proliferation [32]. Other fusion partners for *ALK* include genes such as: *KIF5B*, *KLC1*, *TFG*, *TPR*, *HIP1*, *STRN*, *DCTN1*, *SQSTM1*, *NPM1*, *BCL11A*, and *BIRC6* [33]. Therapeutic options for ALK-positive NSCLC involve *ALK* inhibitors crizotinib (X*ALK*ori, Pfizer), brigatinib (ALUNBRIG, Takeda Pharma), and ceritinib (Zykadia, Novartis Europharm) [8,14]. It was also reported that the use of alectinib (Alecensa, Roche Pharma) (a second-generation *ALK* TKI) as compared to crizotinib has shown a dramatic improvement in PFS and lower toxicities [31].

### 2.3. ROS1-Targeted TKI Treatment

Proto-oncogene receptor tyrosine kinase (*ROS1*) is activated by chromosomal rearrangement in 1–2% cases of NSCLC [8], consisting of point mutations in the *ROS1* gene (e.g., D2033N, G2032R, or L2026M) [34]. Rearrangement results in the fusion of the tyrosine kinase domain of *ROS1* with 1 of 12 different partner proteins. The products of such reactions are constitutively activated, which leads to cellular transformation [35]. The most common *ROS1* fusion partners include *CD74*, *SLC34A2, EZR*, *SDC4*, and *TPM3* [34,35]. In *ROS1*-positive patients treatment with TKI, ceritinib, crizotinib, entrectinib, and lorlatinib are highly effective [8].

### 2.4. BRAF-Targeted TKI Treatment

The serine/threonine-protein kinase (*BRAF*) mutations are found in 3–5% of NSCLC patients [36]. The most commonly reported *BRAF* mutation is an aminoacidic substitution of a valine (V) for a glutamic acid (E) in codon 600 (V600E) [37]. *BRAF V600E* mutations are found in 1–3% of NSCLC and are sensitive to treatment with a combination of *BRAF* inhibitors and dabrafenib (Tafinlar, Novartis Europharm) in combination with trametinib (Mekinist, Novartis Europharm) after progression on chemotherapy [8]. If there is an acquired resistance to *BRAF* inhibitors alone, it is recommended to introduce MEK inhibitors additionally [38].

### 2.5. MET-Targeted TKI Treatment

The proto-oncogene N-methyl-N’-nitroso-guanidine human osteosarcoma transforming gene (*MET*), located in the 7q31 locus of chromosome 7, encodes a receptor tyrosine kinase and induces downstream signaling through the phosphoinositide 3-kinase (PI3K) and RAS-RAF pathways. Abnormal *MET* signaling drives tumor growth through increased cell proliferation, invasion, survival, and metastasis [14,39,40,41]. Several types of *MET* aberrations such as *MET* exon 14 skipping mutation, MET amplification, and *MET* fusions have been observed in numerous different types of cancers. The first target in NSCLC, for which *MET*-targeted therapy was approved in 2020, became *MET* exon 14 skipping mutation (*MET*∆ex 14) [39]. This mutation can result from point mutations, deletions or insertions, or large-scale whole-exon deletions. *MET*∆ex14 occurs in 3 to 4% of patients with NSCLC. In contrast to NSCLC patients with other driver mutations (e.g., *EGFR, ALK*, and *ROS1*), patients with *MET*∆ex14 are over 70 years of age and have a smoking history [40,42,43,44,45,46,47].

Currently, there are many *MET*-TKIs under clinical development. These drugs include selective type 1b inhibitors (e.g., tepotinib, capmatinib, and savolitinib) and nonselective type 1a inhibitors (e.g., crizotinib) [40]. In 2020 and 2021, tepotinib and capmatinib were approved in the USA and Japan, respectively, for use as monotherapies in NSCLC patients carrying *MET* exon 14 skipping [39]. Tepotinib is a highly selective and potent oral *MET* inhibitor that inhibits *MET* phosphorylation and downstream signaling. In preclinical studies, tepotinib inhibited the growth of *MET*-dependent human xenograft tumors and cancer explants [48,49,50,51,52]. Crizotinib, a non-selective type 1a inhibitor, is a multi-tyrosine kinase inhibitor approved for the treatment of advanced NSCLC with *ROS1* or *ALK* rearrangement. In addition to its activity against *ROS1* and *ALK*, crizotinib also exhibits strong activity against *MET* and low nanomolar strength in cell lines that contain changes in the *MET* exon 14 [53].

### 2.6. KRAS-Targeted TKI Treatment

Kirsten rat sarcoma viral oncogene homolog (*KRAS*) is mutated in 15–25% of NSCLC-patients, more frequently than *ALK* rearrangements (~5%) or *MET* mutations (~3%) [54]. *KRAS* mutations have also been found more frequent in non-Asian than in Asian populations (25–50% vs. 5–15%, respectively) [55] and in former or current smokers than in never smokers (25–35% vs. 5%, respectively) [56,57,58,59]. Furthermore, evidence suggested that the presence of *KRAS* mutations combined with *ALK* rearrangements or *EGFR* mutations could negatively impact TKI therapy’s effects [60,61,62,63,64]. The most common *KRAS* mutations occur in codons 12 and 13, including G12C, which is present in 13% of NSCLC.

The first G12C-specific inhibitor able to demonstrate in vivo efficacy was ARS-1620. Since then, other related compounds with increased biological activity have been produced. Sotorasib (AMG-510) and adagrasib (MRTX849) were the earliest of which to enter the clinic [55,65,66]. AMG 510 is a first-in-class oral *KRAS* G12C inhibitor with evidence of clinical activity. Pre-clinical data of AMG510 demonstrated selective targeting of *KRAS* G12C tumors as monotherapy and in combination with cytotoxic therapy [65]. MRTX849 is a potent and selective *KRAS* G12C inhibitor, achieving 65% tumor regression in vitro and in patient-derived xenograft models [66,67].

### 2.7. Testing for Molecular Biomarkers

All mentioned genomic alterations are also known as molecular biomarkers. Molecular testing is used to test for certain biomarkers for available targeted therapies. For eligible patients with locally advanced and resected early-stage NSCLC, molecular testing is also recommended. In 2022 The National Comprehensive Cancer Network^®^ (NCCN^®^) NSCLC Panel added to its Guidelines information about molecular testing, such as a definition for broad molecular profiling for NSCLC. Broad molecular profiling was defined as molecular testing that identifies, e.g., all of the classic actionable biomarkers such as *EGFR, ALK, ROS1, BRAF, MET*, and *KRAS*. Broad genomic profiling can be used to distinguish separate primary lung cancers from intrapulmonary metastases and to assess resistance mechanisms in patients who progressed on targeted therapy. It can also help in determining the eligibility for some molecular clinical trials [10].

Broad molecular profiling systems, such as next-generation sequencing (NGS), may be used to test for multiple biomarkers simultaneously [10]. If the NGS platforms have been designed and validated to detect somatic genomic alterations, they can detect panels of mutations and gene fusions [68,69,70,71,72,73,74,75,76].

If it is clinically possible, molecular testing results for the molecular biomarkers should be known before starting systemic therapy with ICI regimens in eligible patients with advanced NSCLC. If not, then patients are treated as though they do not have driver oncogenes [11,12,31,77].

## 3. Liquid Biopsy

Unfortunately, the quality of the available tumor biopsy and/or cytology material is not always adequate to perform the necessary molecular testing, which has prompted the search for alternatives. The solution may be the use of liquid biopsy, which could impact clinical utility in several ways. Liquid biopsy, as a minimally invasive approach, allows the detection of the disease in peripheral blood samples and it may be useful during cancer screening. Despite some significant efforts that have been attempted so far to develop methods for early cancer detection by using ctDNA presenting promising results, such as CancerSeek [78] and very recently the Galleri Test (Grail) [79], liquid biopsy is not established as a tool yet for early diagnosis. Moreover, liquid biopsy has prognostic value and enables the selection of appropriate therapy and monitoring its efficacy [80,81,82]. Liquid biopsy is based on the detection, isolation, and characterization of circulating tumor cells (CTCs) or/and circulating tumor DNA (ctDNA) or cell-free DNA (cfDNA), exosomes, and tumor-educated platelets (TEP) [83,84].

Over the past few years, great technological advancements have been achieved, and the highly sensitive and specific PCR-based techniques that were used so far for the detection of targeted genomic alterations are gradually replaced by high-throughput NGS techniques. NGS-based methods in liquid biopsy offer a wider spectrum of molecular information obtained through a single analysis. Despite the higher cost, longer turn-around time, and relatively lower sensitivity rates, NGS-based methods in liquid biopsy could positively affect the clinical management of NSCLC patients [85,86]. A multigene NGS approach in liquid biopsy is already included in the guidelines recently issued by the International Society for the Study of Lung Cancer (IASLC) and the European Society for Medical Oncology focused on the personalized treatment of NSCLC patients, providing biomarkers of prognostic significance during disease monitoring, and revealing the presence of alternative druggable alterations at the progression of the disease [87]. Although NGS-based assays are already performed in tissue samples, cfDNA analysis offers some advantages regarding the minimally invasive approach during disease progression and also depicts tumor heterogeneity. Very recently, the FDA approved two NGS liquid biopsy tests, Guardant360 and FoundationOne Liquid CDx, based on the clinical utility of cfDNA testing, for the personalized treatment of NSCLC patients. Many studies based on the NGS approach have been written and are presented in detail below.

### 3.1. Liquid Biopsy Elements

#### 3.1.1. CTCs

CTCs are cells that circulate in the bloodstream after separation from the primary tumor, potentially leading to metastasis formation under favorable conditions. The number of detected CTCs in 10 mL of blood ranges from 0 to over 10,000 [88] but the detection of even single cells may suggest the presence of a developing neoplastic process [89,90,91]. Thus, CTC detection allows for the faster implementation of appropriate treatment, significantly increasing the chances of survival. In most cases, CTC analysis is based on the assessment of the number of cells and their phenotypic characteristics, with some of them enabling also their isolation and subsequent molecular testing [92,93,94,95]. However, all of them carry some challenges. One of the biggest problems may be the high heterogeneity of CTC, observed even in cells derived from the same tumor. The relatively low frequency of CTCs in relation to a large number of blood cells (usually a single CTC per 10^6^–10^7^ leukocytes) may be an additional difficulty [96]. For this reason, the proper analysis should be preceded by the enrichment stage, e.g., increasing the CTC concentration in the analyzed sample [97].

#### 3.1.2. cfDNA

Another source of information about mutations and genetic changes within the tumor is cfDNA. cfDNA is present mainly in the bloodstream, as well as in other body fluids, such as urine, cerebrospinal fluid, and pleural effusions. The analysis of peripheral blood-derived cfDNA can be performed in both serum and plasma. Studies have shown that the concentrations of cfDNA are higher in serum; however, it is not its preferred source due to the potential contamination with genomic DNA [98]. Although cfDNA is rapidly degraded by nucleases [99], it may reflect ongoing genetic changes in the tumor. It has been shown that the amount of cfDNA correlated with the tumor size [100], tumor stage [101], and the presence of metastases [102], suggesting that both primary and secondary genetic changes in the tumor are mirrored in cfDNA. This information is particularly valuable in the aspect of monitoring the course and effectiveness of the applied therapy.

#### 3.1.3. Exosomes

Exosomes are extracellular vesicles ranging in size from 30 to 100 nm. They are detected in the blood of patients with various types of cancer and consist, inter alia, of proteins and nucleic acids. By analyzing molecular changes in their elements (mutations, gene fusions, or splicing variants), it is possible to determine, among others, tumor progression. The double lipid layer of exosomes means it is difficult to purify them, compared to CTCs and cfDNA; however, their structure provides better stability of their content, which allows better identification of the tumor’s origin, genetic changes within it, and potential resistance to treatment. Another advantage of exosomes is their much higher concentration in biological fluids, including blood, than in CTCs and cfDNA [103].

The analysis of isolated exosomes can be performed quickly and accurately with the use of electrochemical and fluorescent technologies, using binding with antibodies, aptamers, or nanomaterials, with various test platforms [103].

#### 3.1.4. TEP

Tumor-educated blood platelets are perhaps the newest ingredients in the liquid biopsy family. The concept of ‘platelet education’ by cancer refers to the presence of specific RNA signatures in platelets from patients with cancer and was first reported in 2010 [104].

Platelets are a fundamental component of the tumor microenvironment and are considered an important aspect of cancer biology as they contribute to tumor initiation, tumor progression, and therapy response. They create an environment supportive of neovascularization, reduce local tumor cell apoptosis and anoikis [105], and are able to induce the epithelial–mesenchymal switch in tumor cells, supporting metastatic spread. The process is generally based on providing mechanical and anti-NK cells protection (transfer of MHC I proteins) of CTCs by cell–fibrin–platelet aggregates [106].

### 3.2. Cancer Screening

An early cancer diagnosis allows for the prompt implementation of an adequate treatment procedure and improves patients’ prognosis. The use of liquid biopsy to detect early-stage cancer is possible but difficult due to the low sensitivity and specificity of the test and the risk of a false-positive result [107,108]. However, the clinical validity of liquid biopsy has been proved, e.g., in lung cancer patients. In a study by Ilie et al., CTCs were detected in 5 out of 168 patients with chronic obstructive pulmonary disease (COPD). Importantly, the detection of CTCs preceded the visualization of lung cancer on computed tomography scans by 1 to 4 years in all CTC-positive patients [82,109]. Another promising study was conducted by Fiorelli et al. using Isolation by Size of Epithelial Tumor Cells (ISET, RareCells, France) filtration technology for CTC enrichment. In this study, CTCs were found in 90% of patients with advanced lung cancer and in 5% with benign lesions, allowing the differentiation of the two types of cancer [81,82,110]. Unfortunately, another attempt with ISET has shown low sensitivity [82,111]. A meta-analysis by Jia et al. confirmed the overall relatively low sensitivity of this approach for the early detection of lung cancer in COPD patients. It also showed that cfDNA shows greater sensitivity and specificity for early cancer detection than ctDNA and CTC and is also the best biomarker for the detection of multiple cancers [112].

### 3.3. Prognostic Value

Numerous studies have shown that the detection of tumor biomarkers through liquid biopsy is a strong prognostic factor. A correlation has been demonstrated between the presence of CTCs, ctDNA, cfDNA, circulating mRNA, and poor PFS and OS in several types of cancer, including patients with melanoma [88,113], lung cancer [114,115], prostate cancer [80,116,117,118], breast cancer [119,120,121,122], and colorectal cancer [123,124,125]. In general, a higher number of CTCs is associated with poor prognosis. In lung cancer, the CTC count is considered as a negative prognostic value that varies depending on the type of tumor. In Small Cell Lung Cancer (SCLC), the detection of more than 50 CTCs in 7.5 mL of blood has a negative prognostic value, while in NSCLC, the detection of 5 CTCs in 7.5 mL of blood and more indicates a poor prognosis [108,126,127]. Not only the number but also the morphological properties of CTCs, such as the ability to form clusters and the presence of apoptotic cells, are important and are associated with a worse prognosis [108,128,129]. The use of CTCs and ctDNA also allows the detection of a minimal residual disease (MRD), and thus the selection of patients more at risk of relapse [108,130,131,132].

### 3.4. Therapy Selection and Monitoring of NSCLC through Liquid Biopsy

In NSCLC, the detection of targetable mutations is required for using the TKI treatment. Those mutations can be detected by the molecular analysis of isolated liquid biopsy elements (Figure 2). In addition, in case of resistance to a given drug, monitoring treatment efficacy using liquid biopsy allows for a timely change of treatment to a more effective one [133]. Liquid biopsy is also used when the patient does not qualify for immunotherapy or targeted therapy due to the lack of specific mutations. Then, if a decision is given about chemotherapy or radiotherapy, it is possible to monitor the CTC counts before and after treatment. Studies have shown that a reduction in CTC counts after chemotherapy is associated with a better prognosis, especially in metastatic breast, colon, and prostate cancer [81,134,135]. There are also a few studies that propose cfDNA concentration as a predictive marker of immunotherapy response [136,137,138].

## 4. Liquid Biopsy Testing in NSCLC

### 4.1. EGFR-Mutant NSCLC

Most of the studies performed so far have focused on ctDNA analysis for *EGFR* mutations. Analysis in CTCs is limited to a few studies, and exosomes even less. Moreover, there are only very few studies that have compared *EGFR* mutations in CTCs and ctDNA directly in the same clinical samples using the same blood draws and the same methodologies. Based on this, we believe that the analysis of *EGFR* mutations in exosomes provides clinically important information that needs to be confirmed through larger clinical studies.

#### 4.1.1. cfDNA Testing

cfDNA analysis in advanced NSCLC has been extensively studied, and its clinical utility is already proven for targeted treatment selection, treatment monitoring, and resistance mechanisms detection [139,140,141]. Thus, current clinical guidelines recommend cfDNA testing for the detection of molecular alterations in NSCLC, either in treatment-naïve patients or in patients who progressed to *EGFR* TKIs [141]. *EGFR-*mutant NSCLC constitutes one of the major paradigms of integrating liquid biopsy testing in the clinical setting that is validated through the liquid biopsy tests already approved by the FDA [142].

The first promising results on the predictive value of cfDNA in NSCLC patients were generated through important clinical trials that compared the efficacy of first-generation *EGFR* TKIs against chemotherapy and included cfDNA analysis, beyond the classic approach of tissue biopsy [143,144]. More precisely, results from the EURTAC trial correlated *EGFR* mutations in cfDNA with OS, PFS, and treatment response and proved the feasibility of using cfDNA instead of tumor biopsy during treatment with erlotinib [145]. High concordance rates between plasma and tissue genotyping for *EGFR* mutations were detected during the ENSURE study, and patients positive for *EGFR* mutations in their plasma had improved PFS when treated with erlotinib compared to chemotherapy [146]. These significant findings led to the approval of the first liquid biopsy test, the cobas *EGFR* Mutation Test (Roche Molecular Systems, Inc., Pleasanton, CA, USA), as a companion diagnostic for erlotinib [147].

NSCLC patients treated with first or second *EGFR* TKIs experience the progression of disease after 9–14 months, mostly because of the presence of the T790M mutation [148]. Therefore, the detection of this resistance mutation is crucial for stratifying patients that could subsequently benefit from treatment with third-generation *EGFR* TKIs [149]. Liquid biopsy analysis, through cfDNA testing, demonstrated a potential role during the progression of the disease. Several studies have shown the concordance between plasma and tissue testing for the detection of T790M. Moreover, noninvasive monitoring captured tumor heterogeneity and identified resistance mutations that otherwise would have been missed while using classical tumor biopsy [150,151,152]. Oxnard et al. also confirmed these results through a retrospective analysis which demonstrated that patients positive for T790M in their plasma were treated with osimertinib and had similar clinical outcomes in terms of PFS compared to the corresponding tissue results [153]. Similar results were observed in the context of the FLAURA trial while evaluating the clinical utility of cfDNA testing to identify patients that could benefit from first-line treatment with osimertinib. Interestingly, in this study, it was shown that in some patients that were positive for *EGFR* mutations in the primary tissue, a lack of *EGFR* mutations in their plasma was observed and this was associated with better PFS; this could be possibly explained as a result of lower tumor burden that was shed in the plasma [154].

The spectrum of resistance mechanisms that occur upon treatment with third-generation *EGFR* TKIs is broadly heterogeneous [155]. The *EGFR* C797S mutation that confers resistance to osimertinib is the most common resistance mechanism detected in plasma [156,157]. Larger cohort studies have revealed various mechanisms of acquired resistance detected in the plasma of NSCLC patients treated with osimertinib either as a second-line treatment [158] or in the first-line setting [159].

The plasma genotyping of *EGFR-*mutant NSCLC is also highly valuable during treatment as an indicator of tumor response or disease progression by evaluating variant allelic fractions of genomic alterations and detecting the emergence of resistance mechanisms earlier. Recently, it was shown that a higher allele frequency of *EGFR* mutations in plasma ctDNA before treatment with osimertinib was a poor prognostic factor [160]. Mok et al. analyzed plasma samples from the FASTACT-2 study to explore the predictive value of changes in the cfDNA *EGFR* mutation status during *EGFR* TKI treatment. They demonstrated that the median PFS and OS were shorter for patients with detectable mutations after three cycles of treatment, underlining the utility of the serial quantitative measurement of *EGFR* mutations in cfDNA to assess tumor progression [161]. Similar results were presented in the Phase III AURA3 trial and in the Phase III FLAURA trial, indicating that the early clearance of ctDNA *EGFR* mutations after 3 or 6 weeks was associated with better clinical outcomes [83,162]. Serial ctDNA monitoring during *EGFR* TKI treatment may be useful for tracking relapse before radiological progression, as was demonstrated by several groups [163,164,165,166].

#### 4.1.2. CTCs

Beyond CTC enumeration, the molecular characterization of CTCs in *EGFR*-mutant NSCLC could offer an alternative source of information critical for clinical decisions. Several groups have detected *EGFR* mutations in CTCs of NSCLC patients with various concordance rates between plasma and/or tissue genotyping. Discrepancies observed between plasma or tissue genotyping and CTC profiling may be attributed to tumor heterogeneity that characterizes NSCLC and also to clonal evolution that occurs under treatment selective pressure [167,168,169,170]. Single CTC analysis for *EGFR* mutation detection in six NSCLC patients revealed intra-patient heterogeneous mutation profiles that reflect rare clones that could lead to therapeutic resistance [171].

Maheswaran et al. successfully identified *EGFR* sensitizing mutations, the resistance mutation T790M, and secondary *EGFR* mutations in CTCs with higher sensitivity than those detected in corresponding plasma samples. Intriguingly, T790M mutation was detected in a majority of patients who had a progression of disease while receiving *EGFR* TKIs [172]. Conversely, in a single-arm phase II clinical trial of erlotinib and pertuzumab, Punnoose et al. observed higher sensitivity for mutation detection in ctDNA than in CTCs [167]. Remarkably, the greatest activity of the pertuzumab-erlotinib combination, with concomitant CTC changes, was seen in patients harboring *EGFR* mutations. Despite that, high CTC counts are usually a poor prognostic factor; in this study, high baseline CTC counts were associated with radiographic response [167]. In a small group of eight metastatic NSCLC patients treated within the LUX-Lung 3 study, Exon 19 deletion *EGFR* mutation was detected by the real-time PCR and melting curve analysis protocol and the results were correlated with radiological response. More specifically, at follow-up, patients without *EGFR-*mutant CTCs relapsed prior to radiological PD whereas patients who had “cleared” CTC showed the significantly prolonged time to treatment failure [173].

Different methodologies have been implemented for the detection of *EGFR* mutations in the CTCs of NSCLC patients, such as the combination of CTC enrichment by the CellSearch (Menarini Silicon Biosystems, Inc., Bologna, Italy) system with next-generation sequencing (NGS), with sensitivity and specificity of 84% and 100%, respectively, corresponding to those present in tumor tissue [174]. Moreover, Gorges et al. explored the feasibility of detecting *EGFR* mutations in CTCs captured by in vivo nanowire CellCollector (GILUPI GmbH, Potsdam, Germany), a procedure that could be useful for treatment monitoring [175].

In addition to the methods already mentioned above, the detection and quantification of *EGFR* mutations in plasma-cfDNA and CTCs of NSCLC patients are feasible thanks to digital PCR (dPCR) technology. Using crystal dPCR and the naica^®^ system (Stilla Technologies), Ntzifa et al. detected *EGFR* mutations in ctDNA and paired CTCs in patients with NSCLC treated with osimertinib at two-time points, before treatment and at the progression of the disease. The results of these studies indicate that the use of crystal dPCR enables the precise, accurate, and highly sensitive detection and quantification of many *EGFR* mutations in plasma-cfDNA and CTC in NSCLC [169].

#### 4.1.3. Exosomes

During the last few years, the analysis of circulating exosomes has provided new opportunities for cancer diagnosis and the monitoring of disease progression in the liquid biopsy field. Many studies on NSCLC have shown higher sensitivity and specificity while combining exosomes with cfDNA testing, thus implicating their clinical utility” [176]. In a small study including 41 NSCLC patients, it was demonstrated that patients with mutated plasma exoNA (*KRAS, EGFR, BRAF*) and low MAF had longer median PFS and time-to-treatment failure, suggesting that the molecular profiling of plasma exoNA can be predictive of clinical outcomes [177].

Castellanos-Rizaldos et al. successfully identified T790M-positive NSCLC patients through exoNA analysis and they proved that the combination of exoRNA/DNA and cfDNA detection offers higher sensitivity and specificity than using cfDNA alone [176]. Later, the same group developed and validated a qPCR-based test that detects a panel of 29 *EGFR* mutations in exosomal RNA/DNA and cfDNA that predict the response to first-line *EGFR* TKIs and osimertinib [84]. The increased sensitivity for *EGFR* mutation detection by combining exoRNA/DNA and cfDNA analysis was also confirmed by Krug et al. [178].

Interestingly, the exosome-based detection of *EGFR* T790M in the plasma and pleural fluid of prospectively enrolled NSCLC patients after first-line TKI therapy also demonstrated greater sensitivity [179]. Furthermore, the longitudinal *EGFR* mutation analysis of bronchial washing (BW)-derived extracellular vesicles (EVs) revealed an excellent correlation with disease progression, as measured by CT images [180].

Recently, in a small pilot study with 10 metastatic *EGFR-*mutant NSCLC patients, it was shown that EVs had a better detection rate that ctDNA, and that variations in the mutant EV-RNA burden could mirror disease status [181].

Exosomes carry important molecular information since they are secreted from living cells, and in some cases, their analysis could be more informative than cfDNA [182]. However, there are still challenges to overcome, and further research needs to be undertaken in order to integrate exosome analysis into the liquid biopsy setting

### 4.2. ALK-Rearranged NSCLC

#### 4.2.1. ctDNA

Different NGS methodologies, such as amplicon-based ctDNA NGS [183,184] or hybrid-capture techniques [185,186,187,188,189,190], have been mostly used in several studies for the detection of *ALK* rearrangements in newly diagnosed NSCLC patients or those who relapsed after targeted treatment. However, recently, Dietz et al. combined targeted NGS with copy number variation profiling using the shallow whole genome sequencing (sWGS) of ctDNA in order to improve the longitudinal monitoring of *ALK*-positive NSCLC patients in particular for cases without detectable mutations in ctDNA or with a wide range of acquired genomic alterations during therapy [191].

In a recent study, plasma samples were collected and analyzed after progression on first, second, or third-generation *ALK* TKIs treatments. These patients were sequentially treated with different *ALK* inhibitors and longitudinally monitored through the course of their treatment; it was shown that *ALK* mutations emerged as a result of increased lines of *ALK* inhibitors, thus indicating the importance of plasma genotyping during treatment in order to guide clinical decisions [192].

Many other studies are in line with the idea of detecting resistance mechanisms to *ALK* inhibitors through plasma genotyping, highlighting the advantages of liquid biopsy: high concordance rates with tissue biopsy, shorter turnaround times, and, most importantly, the potential of ctDNA analysis to efficiently guide treatment decisions in case tissue biopsy is negative or not feasible [185,193,194,195,196]. Furthermore, numerous studies have demonstrated the heterogeneous spectrum of *ALK* rearrangements observed in ctDNA as resistance mechanisms and its association with treatment outcomes [195]. The most common *ALK* fusion detected is *EML4*-*ALK*, leaving aside other partner fusions such as *STRN-ALK* [188], and less frequent *KCNQ*, *KLC1*, *KIF5B*, *PPM1B*, *TGF* [188], and *PON1-ALK* [193]. *ALK* mutations reported so far are G1202R, L1196M, F1174X, G1269A, and I1171X [185,186,192,196,197]. Interestingly, Shaw et al. examined the efficacy of lorlatinib (Lorviqua Pfizer) according to plasma or tissue genotyping and concluded that mutation-positive patients had significantly higher response rates to lorlatinib compared to mutation-negative patients. However, PFS did not differ significantly in patients with and without *ALK* mutations [198]. In the ongoing randomized phase III CROWN study, it was shown that a decrease in the mean variant allele fraction (VAF) of *ALK* alterations (fusions and/or mutations) 4 weeks after lorlatinib in *ALK*-positive NSCLC, previously untreated, may be associated with better responses and longer PFS. These important data support the clinical utility of early ctDNA dynamics during treatment with *ALK* inhibitors [199]. In addition to the initial results of the phase III ALEX study, a retrospective analysis based on plasma analysis suggested that patients treated with alectinib (Alecensa, Roche Pharma) had longer PFS compared to those treated with crizotinib (XALKori, Pfizer), irrespective of the *EML4-ALK* variant [31]. Moreover, recent data retrieved from the same study showed the clinical utility of cfDNA concentrations during treatment and its correlation with treatment outcomes depending on *ALK* inhibitors [115]. The BFAST study presented comparable results to the ALEX study regarding the ORR of alectinib and was the first trial to use a plasma-based NGS method as a sole approach to identify genomic alterations that could aid clinical decision-making for patients with *ALK*-positive NSCLC [187].

The eXalt2 study was the first study to assess the clinical utility of analyzing ctDNA as a function of the response to ensartinib (X-396). The differences observed in the response rates were attributed to specific variants of *EML4-ALK* fusion detected [186]. In a multicenter phase 2 trial, in patients with *ALK*-positive NSCLC who progressed on crizotinib and were subsequently treated with ensartinib, longitudinal ctDNA analysis revealed *ALK*-dependent (G1269A, G1202R, and E1210K mutations) and *ALK*-independent (TP53 mutation) resistance mechanisms, thus underlining the significance of ctDNA analysis for monitoring tumor evolution [197].

In cases of central nervous system (CNS) metastases in NSCLC (brain or leptomeningeal), plasma ctDNA genotyping is constrained by a blood-tumor barrier, and thus alternative sources of ctDNA, such as cerebrospinal fluid, can yield a higher source of ctDNA for the detection of *ALK* rearrangements [200,201].

#### 4.2.2. CTCs

One of the first studies that explored the feasibility of detecting *ALK* rearrangements in CTCs by FISH and ICC included 87 lung adenocarcinoma patients. Only five of them were found positive for *ALK* rearrangements, and the results were in concordance with tissue genotyping. Unlike tumor cells, all CTCs were found to be positive for *ALK* rearrangements, demonstrating an aggressive type of tumor cells [202].

Pailler et al. detected *ALK* rearrangements in CTCs of 18 *ALK*-positive NSCLC patients by FA-FISH with a cutoff of ≥ 4 CTCs/1mL of peripheral blood. Surprisingly, it was observed that *ALK*-rearranged CTCs expressed a mesenchymal phenotype, whereas all tumors had a more heterogeneous profile suggesting the invasiveness of CTCs promoted through EMT. *ALK*-rearranged CTC levels during the treatment monitoring of five patients with crizotinib presented different response patterns [203]. Moreover, in another small study, it was also shown that CTCs recapitulate the *ALK* rearrangement status of tumor tissue, and, therefore, CTCs represent a suitable alternative to tissue biopsy for guiding treatment [204]. In a study of 39 *ALK*-rearranged NSCLC patients treated with crizotinib, Pailler et al. found aberrant *ALK* copy number gain in CTCs and correlated the dynamic changes in the levels of these CTCs with PFS. Therefore, the longitudinal monitoring of these patients through CTC analysis proved to be a promising tool for clinical outcome prediction [205].

In case study reports of *EML4-ALK*-positive NSCLC patients who underwent sequential monitoring during therapy with *ALK* inhibitors, it was found that *EML4-ALK*-positive CTCs reflect the response to these inhibitors and predict treatment resistance [206,207]. Recently, in a larger study including 203 stage IIIB/IV NSCLC patients, it was shown the complementary value of detecting *ALK*-rearranged CTCs during treatment with *ALK* inhibitors through serial blood sampling. At baseline, there was a high concordance between tissue and CTC analysis. However, no significant association was observed between CTC levels and OS or PFS [208].

CTC analysis at the single-cell level revealed acquired resistance mechanisms during treatment with *ALK* inhibitors. Interestingly, in a patient resistant to lorlatinib *ALK* compound mutations were detected in two single CTCs and only one of them was present in the corresponding tumor biopsy. According to these results, tumor heterogeneity can be reflected in CTCs, and single-cell analysis can guide personalized treatment options [209]. In a recent exploratory study conducted by the same group, single CTC analysis from six *ALK*-rearranged patients resistant to crizotinib or lorlatinib showed aberrant CNA profiles and high levels of chromosomal instability at resistance, suggesting that CTCs portray the heterogeneous pattern of drug resistance to *ALK*-TKIs [210].

#### 4.2.3. Exosomes and TEPs

Alternative liquid biopsy components for longitudinal monitoring during treatment with *ALK* inhibitors could also be exosomes and platelets that seem to carry useful molecular information related to tumors. In a prospective cohort of *ALK*-positive patients, it was feasible to detect *EML4-ALK* rearrangements in plasma exosomes of 50% of these patients using qPCR, whereas the detection of *ALK* mutations at PD was correlated to poor response to treatment [211]. In a similar study, 9 out of 14 patients with a confirmed diagnosis of stage IIIB–IV NSCLC, naïve or under treatment with a known *ALK* status, were found positive for *EML4-ALK* in RNA isolated from exosomes using NGS [212]. Moreover, platelets released by tumor cells (tumor-educated platelets, TEPs) could effectively mirror the clinical status of NSCLC patients under crizotinib treatment. In a group of 29 NSCLC patients, *EML4-ALK*-rearrangements were detected in platelets and correlated with shorter PFS, whereas the serial monitoring of one patient revealed resistance to crizotinib prior to radiographic PD based on *EML4-ALK*-positive platelets [213]. Contrary to these results, Park et al. found that patients positive for *EML4-ALK* fusions presented longer median durations of treatment, PFS, and higher ORR [214]. Nevertheless, platelets are a valuable alternative source for the detection of *ALK* rearrangements and further investigation needs to be undertaken for their clinical utility.

### 4.3. ROS1-Positive NSCLC

#### 4.3.1. ctDNA

As mentioned above, *ROS1* fusions are of low prevalence in NSCLC (1–2%), and as a result, studies including exclusively this subset of NSCLC patients are missing. Limited but significant studies on ctDNA analysis have shown some important results regarding the detection of *ROS1* rearrangements in plasma as a tissue surrogate. Most of the studies included NGS panels for the most important therapeutically targetable mutations for the appropriate TKI therapy in NSCLC patients or treatment monitoring [215,216,217] and also proved the clinical utility of cfDNA testing at diagnosis and the potential it offers for faster, minimally invasive therapeutic decisions [218]

NGS techniques, such as amplicon-based plasma NGS, were previously used to detect *ROS1* rearrangements in the plasma of NSCLC patients with known targetable genotypes, either before initiating targeted therapy or during treatment. Guibert et al. had successfully identified *CD74-ROS1* rearrangements in two patients [183], whereas Mezquita et al. tried to assess the clinical utility of *ROS1* fusion and resistance mutations using an amplicon-based liquid biopsy test in NSCLC patients. They observed that 30% of cases harbored *ROS1* resistance mutations and experienced rapid progression of disease compared to those who had undetectable mutations. Only in one case has *ROS1* mutation emerged during crizotinib failure [184].

A large study of NGS-based genotyping, with the hybrid-capture-based Guardant360x assay, in *ROS1*-positive NSCLC patients revealed high concordance (100% for 7 patients from 54) between plasma and corresponding tissue samples. Among the spectrum of *ROS1* fusions that were found, *CD74-ROS1* fusion was the most prevalent [219]. Plasma samples were analyzed for a smaller group of patients at relapse on crizotinib and it was found that the sensitivity for detecting *ROS1* rearrangements was 50%. In post-crizotinib plasma samples, 33% of patients harbored the *ROS1* G2032R mutation, whereas the remainder harbored the L2026M gatekeeper mutation. Among them, one patient was not found to have the same mutation also in tissue biopsy, but only after the resistance with chemotherapy and crizotinib, there was a concordance. These results may be indicative of spatial tumor heterogeneity and emphasize the importance of plasma genotyping during the treatment monitoring of these patients [219].

In the context of a phase I/II trial for the efficacy of lorlatinib (Lorviqua, Pfizer) in *ROS1*-positive NSCLC patients, it was found that before treatment with lorlatinib 15% of patients previously treated with crizotinib had *ROS1* mutations in ctDNA (G2032R, L2026M, L2026M, and I2025I), whereas TKI treatment-naïve patients had no detectable *ROS1* mutations [220]. However, patients harboring the most common *ROS1* mutation, G2032R, did not achieve a response to lorlatinib in contrast to other types of mutations. Therefore, ctDNA genotyping before the initiation of lorlatinib is a critical step for assessing the efficacy of treatment [220].

Very recently, the FoundationOne Liquid CDx test was evaluated for its clinical validity for the identification of patients with *NTRK* or *ROS1* fusions that may benefit from treatment with entrectinib (Rozlytrek, Roche Pharma) or those with acquired resistance to TKIs. It was found that plasma testing can be used as a complement to tissue genotyping for clinical decisions [221]. Another larger study in a pan-cancer patient population (36,916 ctDNA samples and 368,931 tumor tissue samples) confirmed that ctDNA analysis could reliably identify oncogenic fusions with high concordance to tissue genotyping results [222].

#### 4.3.2. CTCs

The first attempt to detect *ROS1* rearrangements in the CTCs isolated by ISET technology of four NSCLC patients under crizotinib was undertaken by Pailler et al., who used a filter-adapted-fluorescence in situ hybridization (FA-FISH) protocol. They further associated variations in *ROS1*-rearranged CTC levels with clinical evolution in three of them and also found differences in copy numbers compared with tumor biopsy [223]. In another small study, *ROS1* rearrangements were detected in enriched CTCs also by FISH, confirming, therefore, that CTCs can offer a reliable alternative for the detection of *ROS1*-rearrangements in NSCLC patients [224]. Finally, the clinical utility of the detection of *ROS1* rearrangements in CTCs was shown in a case report study for a patient with peritoneal carcinomatosis. The patient benefited from crizotinib and ceritinib treatment in the long term based on the detection of *ROS1* rearrangements in CTCs by FISH analysis, whereas NGS plasma genotyping was negative [225].

### 4.4. BRAF Mutated NSCLC

#### ctDNA and CTCs

According to NCCN guidelines, *BRAF* mutation testing is one of the nine recommended molecular biomarkers to be tested in newly diagnosed metastatic NSCLC patients. Based on that, the NILE study approved that comprehensive cfDNA testing for these biomarkers is non-inferior compared to tissue genotyping [226]. However, there is limited evidence for testing *BRAF* mutations in the plasma of NSCLC patients according to real-world experimental data. As a result, there is still a need to investigate its role in guiding treatment to these patients [227]. In a retrospective analysis, *BRAF* mutation testing was conducted in plasma cfDNA samples in cases of tissue unavailability, and a small but important percentage of patients harbored *BRAF* V600E [227]. Therefore, it was highlighted that cfDNA testing as a potential alternative could aid therapeutic decisions and successfully indicate those patients that may benefit from a *BRAF* TKI treatment [227]. Another crucial aspect of cfDNA testing for *BRAF* mutations is the longitudinal monitoring of NSCLC patients during *BRAF*-targeted treatment [228]. Early plasma dynamics of *BRAF* mutations proved to be a reliable predictor of response to treatment with *BRAF* inhibitors, and also plasma genotyping provided information about resistance mechanisms [228]. Furthermore, there have also been reported some non-negligible case studies for patients that have benefited from combinational *EGFR-* and *BRAF-*targeted treatment based on cfDNA testing during their disease monitoring [229]. For instance, Solassol et al. reported a successful treatment management of an NSCLC patient administering dabrafenib plus trametinib and osimertinib based on sequential liquid biopsy testing [230]. Two other similar case studies have been reported, where patients positive for *EGFR* and *BRAF* mutations in plasma cfDNA responded well to the concurrent combination of dabrafenib, trametinib, and osimertinib [231,232]. Besides *BRAF* V600 mutations, 50–80% of *BRAF* mutations are non-V600 and are classified into two categories based on their kinase activity and sensitivity to RTK inhibitors [233]. The SLLIP trial retrospectively assessed the *BRAF* mutational spectrum of 185 newly diagnosed advanced lung adenocarcinoma patients and indicated that the class of *BRAF* mutations should be taken into account during therapeutic decisions, as this was confirmed through cell viability experiments [233]. To date, there is only one short report about detecting *BRAF* mutations in cfDNA and CTCs in six patients during treatment with dabrafenib and trametinib. Plasma cfDNA was found to have better sensitivity at detecting and monitoring mutations compared with CTCs [234]. Nevertheless, this must be investigated through larger cohort studies.

### 4.5. MET exon14-Positive NSCLC

#### ctDNA

The *MET*∆14 mutation in NSCLC patients can be detected through cfDNA or RNA from the patient’s plasma using either DNA sequencing based on NGS. Palik et al. used this form of liquid biopsy in their studies to determine the effectiveness of treatment in patients with NSCLC. In their open-label Phase 2 study, tepotinib was administered once daily to patients with advanced or metastatic NSCLC with a confirmed exon 14 *MET* skipping mutation. The response to the treatment was analyzed, inter alia, by detecting the presence of an exon 14 *MET* skipping mutation using liquid biopsy or tissue biopsy. The plasma cfDNA was analyzed with the use of NGS panel Guardant360 (which includes 73 genes). The plasma was collected at baseline, at weeks 6 and 12, and at the end of treatment. The authors noted that, although the use of the molecular cfDNA response is not yet part of standard practice in the treatment of solid tumors, correlations between changes in cfDNA levels and tumor response have been reported in several types of cancer, including lung cancer. It was found that the baseline cfDNA analysis performed provided valuable insight into the mutation profiles of patients with exon 14 *MET* skipping mutations [40].

The problem of resistance to targeted therapy in NSCLC patients also applies to those with the exon 14 *MET* skipping mutation. Sai-Hong Ignatius Ou et al. tried to describe the full spectrum of resistance mechanisms to crizotinib in *MET*ex14-positive NSCLC. In order to evaluate the potential mechanisms of resistance, the ctDNA of patients with *MET*ex14-positive NSCLC enrolled in treatment with crizotinib was analyzed. The isolated ctDNA was analyzed using an Illumina HiSeq 2500 sequencer (Illumina, San Diego, CA, USA). Based on the obtained results, it was concluded that the appearance of the preexisting *MET* Y1230C likely causes resistance to crizotinib, in this case, of *MET*ex14-positive NSCLC. Additionally, the study showed that non-invasive ctDNA assays can be a convenient method of detecting resistance mutations in patients with previously known targeting mutations [235].

### 4.6. KRAS G12C Mutated NSCLC

#### ctDNA

In advanced-stage NSCLC patients, liquid biopsy may be an option for *KRAS* testing to select patients for TKI treatment [236]. *KRAS* G12C can be detected through ctDNA from the patients’ plasma using DNA sequencing based on NGS and/or specifically a real-time polymerase chain reaction (PCR). Nicolazzo et al. compared those two assays to track the *KRAS* G12C mutation at the onset of progression from previous lines of therapy. Plasma samples were collected from 38 NSCLC patients with radiologically confirmed disease progression on any first-line treatment (checkpoint inhibitors/platinum-based doublet chemotherapy/targeted therapy). All the plasma samples were obtained at the Time of Disease Progression from the first treatment line. All the plasma samples obtained at PD were first screened for the *KRAS* G12C mutation through real-time PCR (IdyllaTM, Biocartis, Jersey City, NJ, USA). The *KRAS* G12C mutation was detected in 24% of the ctDNA samples. The results obtained through Idylla were confirmed in 100% of the cases by analysis of the plasma samples through NGS. It was also shown that *KRAS* G12C coexisted with *EGFR* mutations in two cases; p53 in two cases; *MAP2K1* in two cases; *PIK3A* in one case; *BRAF* in one case. G12C cooccurred with other *KRAS* mutations in one case. This pilot study may suggest that in the assessment of the plasma, with *KRAS* G12C mutation as a druggable target, a real-time PCR assay through Idylla might be a suitable approach to better match patients to interventional biomarker-targeted therapies [237].

Plasma ctDNA can also be analyzed with the use of the NGS panel Guardant360. The Guardant360 assay can analyze point mutations in 54–74 genes, copy number amplifications in up to 18 genes, and fusions in up to 6 genes. Thein et al. performed a comprehensive analysis of *KRAS* G12C mutations in solid tumors. *KRAS* G12C mutations were identified in 2985 of 80,911 patients (3.7%) across > 40 tumor types, as detected by circulating tumor DNA. *KRAS* G12C mutations were detected most frequently in patients with nonsquamous non-small-cell lung cancer (NSCLC; 7.5%). They were also detected in patients with NSCLC of all subtypes (6.9%), cancer of unknown primary (4.1%), colorectal cancer (3.5%), squamous NSCLC (2.0%), pulmonary neuroendocrine tumors (1.9%), and pancreatic ductal cholangiocarcinoma (1.2%) and adenocarcinoma (1.2%). Thein et al. also found a very high positive predictive value between tissue and liquid biopsies performed within 6 months of each, whereas between tests conducted > 6 months apart, the positive predictive value was lower at 77%. The study had one critical limitation. According to the authors, the Guardant360 database includes both treatment-naïve and previously treated patients without the necessary details to analyze them separately. It is limiting the ability to compare detection rates from this study with the prevalence rates previously published. Despite this, the study shows that the Guardant360 assay is a good tool to identify *KRAS* G12C mutations in ctDNA [238].

As an example, there was a case report of a patient with advanced NSCLC who was initially started on Alectinib based on positivity for *ALK* gene rearrangement found in the FISH study. After 2 months, the progression of the disease and new osseous were shown by an interim PET scan. At that time, a liquid biopsy was obtained, and cell-free DNA was tested via Guardant360. Instead of *ALK* rearrangement, *KRAS*-pG12C was detected, and the patient’s treatment was changed to sotorasib, just after its FDA approval [239].

## 5. Conclusions

Due to the high mortality rate of lung cancer patients, the efforts of cancer researchers are now focused on increasing the use of molecular targeted therapy and the improvement of its monitoring methods. Based on the presented research, we believe that liquid biopsy is an increasingly popular predictive tool in the treatment of NSCLC based on TKI. The studies presented in this article show that detection and analysis based on CTC, cfDNA, exosomes, and/or TEP can contribute to the appropriate selection and monitoring of targeted therapy in NSCLC patients as complementary to invasive tissue biopsy. The detection of these elements, combined with their molecular analysis, enables the detection of targetable mutations, which are required for TKI treatment. Mutations (*EGFR, ALK, ROS1, BRAF, MET, KRAS*) detected in liquid biopsy elements using molecular detection methods (e.g., dPCR, NGS, sWGS) correspond to the results obtained during traditional tissue biopsy. It is worth mentioning that tissue biopsy may be detrimental to the patient’s health due to its invasiveness. Additionally, the quality of the available tumor biopsy and/or cytology material is not always adequate to perform the necessary molecular testing. Therefore, liquid biopsy can be a competitive predictive tool for tissue biopsy. Despite such promising results obtained by many research teams, we think that it is still necessary to carry out prospective studies on a larger group of patients to validate these methods before their application in clinical practice.

## Figures and Tables

**Figure 1 cells-11-02871-f001:**
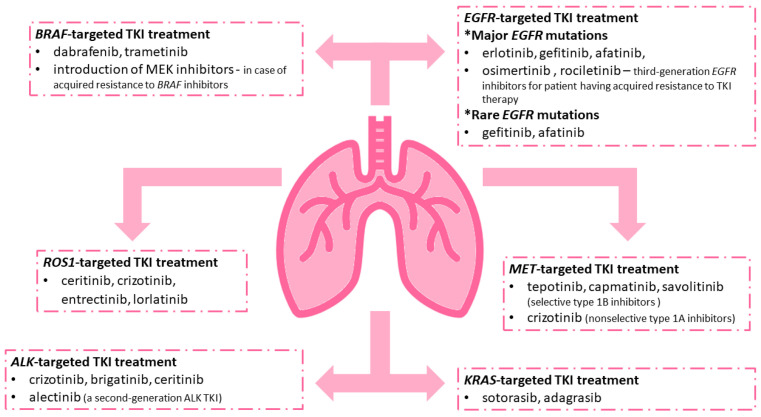
Examples of tyrosine kinase inhibitors (TKIs) used in NSCLC patients with certain changes in genes including the *EGFR*, *ALK*, *ROS1*, *BRAF*, *MET*, and *KRAS* genes.

**Figure 2 cells-11-02871-f002:**
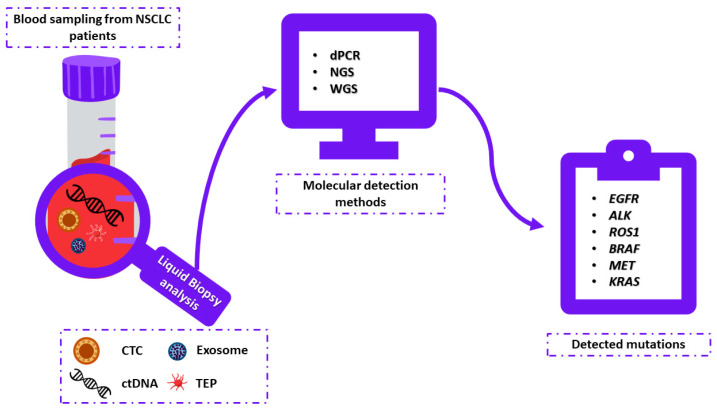
Liquid biopsy analysis as a diagnostic tool for TKI-based treatment in non-small cell lung cancer. In peripheral blood samples, liquid biopsy elements such as circulating tumor cells (CTCs) and/or cell-free DNA (cfDNA) and/or tumor-educated platelets (TEPs) and/or exosomes can be detected. Mutations (*EGFR*, *ALK*, *ROS1*, *BRAF*, *MET, KRAS*) detected in liquid biopsy elements using molecular detection methods (e.g., next generation sequencing (NGS), digital PCR (dPCR), shallow whole genome sequencing (sWGS)) correspond to the results obtained during traditional tissue biopsy.

## Data Availability

Not applicable.

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
