# Peer review of "Liquid Biopsy Analysis as a Tool for TKI-Based Treatment in Non-Small Cell Lung Cancer"

_cells, 2022, doi:10.3390/cells11182871_

Round 1

Reviewer 1 Report

The manuscript entitled:" Liquid biopsy analysis as a tool for TKI-based treatment in 2 non-small cell lung cancer" focuses on a systemic reviison of literature data about the role of liquid biopsy in the clinical managment of lung cancer patients. Accordingly, the manuscript is technically correct but requires some moderate integrations to be accepted for the publication.

- In the introduction section, please, could the authors could the authros underline the aim of this review?

- In the therapeutic approaches, some limitations should be discussed. In my opinion, the authors should report the clinical stage where TKIs are approved. In addition, a dedictaed section where technical strategies available to analyze clinically relevant biomarkers should represent a point of strenght for the manuscript.

- In the paragraph. 2.1, please, could the authors also discuss about the role of "uncommon EGFR" mutations detectable in diagnostic routine? How these molecular laterations may impact on the guidance of NSCLC patients?

- In the therapeutic section, a deep investigationb of clinical role of these biomarkers should be carried out by the authors.

- In the paragraph about liquid biopsy tool, please, could the authors also discuss how KRAS p. G12C alteration could be detected in this setting?

- In the liquid biopsy section, an extensive analysis of large NGS panels evaluated on liquid biopsy specimens should be carried out

- Section 3 appears too short. In my opinion, othert relevant literature data should be implemented to clarify the integrating analytes found in torrent blood .

Author Response

Response to Reviewer 1

We would like to thank the Reviewer for the insightful and very constructive comments and suggestions that have guided us to significantly improve the quality of our review. We have revised the manuscript according to the comments and you will find below a point-by-point description of our modifications.

  1. In the introduction section, please, could the authors the authors underline the aim of this review?

RE: Corrected according to Reviewer’s suggestion (page 1, lines 24-28).

  1. In the therapeutic approaches, some limitations should be discussed. In my opinion, the authors should report the clinical stage where TKIs are approved. In addition, a dedicated section where technical strategies available to analyze clinically relevant biomarkers should represent a point of strength for the manuscript.

RE: We followed the reviewer suggestion. The information has been added to the “Therapeutic approaches in NSCLC” section: “TKIs are used in patients with metastatic NSCLC (stage III or IV) and certain gene alterations, including EGFR, ALK, ROS1, NTRK1, BRAF, HER2, MET and KRAS genes, which are independent of PD-L1 levels. Patients with metastatic NSCLC, targetable driver oncogene molecular variant and also PD-L1 expression levels of 1% or more should receive first-line targeted therapy for detected oncogene instead of first-line immune checkpoint inhibitors (ICIs). Targeted therapies yield higher response rates (e.g., osimertinib, 80%) than ICIs (poor response rates) in the first-line setting and are better tolerated [10–13].” (page 2, lines 74-81).

  1. In the paragraph. 2.1, please, could the authors also discuss about the role of "uncommon EGFR" mutations detectable in diagnostic routine? How may these molecular alterations impact on the guidance of NSCLC patients?

RE: We followed the reviewer suggestion. The information about the uncommon EGFR mutations has been added to the new “Rare EGFR mutations” subsection:
“Despite the fact, that the most common EGFR mutations in NSCLC are L858R substitution and exon 19 deletion, 10% of NSCLC patients will have an uncommon EGFR mutation. Generally, mutations involving exons 18 to 21 are considered sensitive to EGFR TKIs, with the exception being mutations involving exon 20, including T790M and exon 20 insertions [19]. Mutations in exon 18 are typically considered sensitizing to EGFR TKI therapy. The most frequently detected exon 18 mutation is the G719X mutation [23,24]. Compared with wild-type EGFR,53, the G719X mutation is associated with a 10-fold in-crease in EGFR activation. However, in vitro studies have suggested that it is not as sensitive to gefitinib as NSCLC cell lines with L858R mutations [25]. Patients with G719X mutations respond to EGFR TKI therapy have not always been as prolonged as those seen with the more common mutations. However, complex mutations within exon 18 (e.g., L861Q mutation) may be associated with a better prognosis than point mutations [24,26]. Yang et al. study, in which 12% of patients had L861Q substitutions, has shown the association of the presence of the L861Q mutation with sensitivity to EGFR TKIs [18,27]. For those patients the objective response rate (ORR) was 56.3% with a median progression-free survival (PFS) of 8.2 months and overall survival (OS) of 17.1 months [28]. Another mutation, S768I mutation, in some studies [29,30] is associated with poor responses to TKIs. However, patients with S768I mutation, from treatment with afatinib did show good clinical outcomes, with a median PFS of 14.7 months and median OS not yet reported (95% CI: 3.4 months–not estimable) [28].” (pages 3-4, lines 119-141)

  1. In the therapeutic section, a deep investigation of clinical role of these biomarkers should be carried out by the authors.

RE: We followed the reviewer suggestion. The information about the clinical role of molecular biomarkers has been added to the “Therapeutic approaches in NSCLC” section:
“All mentioned genomic alterations are also known as molecular biomarkers. Molecular testing is used to test for certain biomarkers for available targeted therapies. For eligible patients with locally advanced and resected early-stage NSCLC molecular testing is also recommended. In 2022 The National Comprehensive Cancer Network® (NCCN®) NSCLC Panel added to its Guidelines information about molecular testing, such as a definition for broad molecular profiling for NSCLC. Broad molecular profiling was defined as molecular testing that identifies e.g., all of the classic actionable biomarkers such as EGFR, ALK, ROS1, BRAF, MET and KRAS. Broad genomic profiling can be used to distinguish separate primary lung cancers from intrapulmonary metastases and to assess resistance mechanisms in patients who progressed on targeted therapy. It can also help in determining the eligibility for some molecular clinical trials [10]. Broad molecular profiling systems, such as next-generation sequencing (NGS), may be used to test for multiple biomarkers simultaneously [10]. If the NGS platforms have been designed and validated to detect somatic genomic alterations, they can detect panels of mutations and gene fusions [68–76]. If it is clinically possible molecular testing results for the molecular biomarkers should be known before starting systemic therapy with ICI regimens in eligible patients with advanced NSCLC. If not, then patients are treated as though they do not have driver oncogenes [11,12,31,77].” (pages 5-6, lines 219-238).

  1. In the paragraph about liquid biopsy tool, please, could the authors also discuss how KRAS p. G12C alteration could be detected in this setting?

RE: We followed the reviewer suggestion and added information about how KRAS p. G12C alteration could be detected by liquid biopsy tools in “KRAS G12C mutated NSCLC. ctDNA” subsection:
“In advanced-stage NSCLC patients, liquid biopsy may be an option for KRAS testing to select patients for TKI treatment [238]. KRAS G12C can be detected through ctDNA from the patient’s plasma using DNA sequencing based on NGS and/or specifically real-time polymerase chain reaction (PCR). Nicolazzo et al. compared those two assays to track the KRAS G12C mutation at the onset of progression from previous lines of therapy. Plasma samples were collected from 38 NSCLC patients with radio-logically confirmed disease progression on any first-line treatment (checkpoint inhibitors/platinum-based doublet chemotherapy/targeted therapy). All the plasma samples were obtained at the Time of Disease Progression from the first treatment line. All the plasma samples obtained at PD were first screened for the KRAS G12C mutation through the real-time PCR (IdyllaTM, Biocartis, Jersey City, NJ, USA). The KRAS G12C mutation was detected in 24% ctDNA samples. The results obtained through Idylla were confirmed in 100% of the cases by analysis of the plasma samples through NGS. It was also shown that KRAS G12C coexisted with EGFR mutations in two cases; p53 in two cases; MAP2K1 in two cases; PIK3A in one case; BRAF in one case. G12C cooccurred with other KRAS mutations in one case. This pilot study may suggest that in the assessment of the plasma KRAS G12C mutation as a druggable target, real-time PCR assay Idylla might be a suitable approach to better match patients to interventional biomarker-targeted therapies [239]. Plasma ctDNA can also be analyzed with the use of NGS panel Guardant360. The Guardant360 assay can analyze point mutations in 54-74 genes, copy number amplifications in up to 18 genes, and fusions in up to six genes. Thein et al. performed a com-prehensive analysis of KRAS G12C mutations in solid tumors. KRAS G12C mutations were identified in 2,985 of 80,911 patients (3.7%), across > 40 tumor types, as detected by circulating tumor DNA. KRAS G12C mutations were detected most frequently in patients with nonsquamous non–small-cell lung cancer (NSCLC; 7.5%). They were also detected in patients with NSCLC of all subtypes (6.9%), cancer of unknown primary (4.1%), colorectal cancer (3.5%), squamous NSCLC (2.0%), pulmonary neuroendocrine tumors (1.9%), and pancreatic ductal cholangiocarcinoma (1.2%) and adenocarcinoma (1.2%). Thein et al. also found a very high positive predictive value between tissue and liquid biopsies performed within 6 months of each. Whereas between tests conducted > 6 months apart the positive predictive value was lower at 77%. The study had one critical limitation. According to the authors, the Guardant360 database includes both treatment-naive and previously treated patients without the necessary details to analyze separately. It is limiting the ability to compare detection rates from this study with prevalence rates previously published. Despite this, the study shows, that Guar-dant360 assay is a good tool to identify KRAS G12C mutations in ctDNA[240]. As an example, there is a case report of advanced NSCLC who was initially start-ed on Alectinib based on positivity for ALK gene rearrangement found in the FISH study. After 2 months progression of the disease and new osseous have been shown by interim PET scan. At that time liquid biopsy was obtained and cell-free DNA was test-ed via Guardant360. Instead of ALK rearrangement, KRAS-pG12C was detected, and the patient’s treatment was changed on sotorasib, just after its FDA approval [241]” (pages 16-17, lines 765-808).

We also added information’s about KRAS mutations and treatment methods:
“Kirsten rat sarcoma viral oncogene homolog (KRAS) is mutated in 15-25% of NSCLC-patients, more frequently than ALK rearrangements (~5%) or MET mutations (~3%) [54]. KRAS mutations have also been found more frequent in non‐Asian than in Asian populations (25–50% vs 5–15%, respectively) [55] and in former or current smokers than in never smokers (25-35% vs 5% respectively) [56–59]. Furthermore, evidence suggested that the presence of KRAS mutations combined with ALK rearrangements or EGFR mutations could negatively impact TKI therapy's effects [60–64]. The most common KRAS mutations occur in codons 12 and 13, including G12C, which is present in 13% of NSCLC. The first G12C-specific inhibitor able to demonstrate in vivo efficacy was ARS-1620. Since then, other related compounds with increased biological activity have been produced. Sotorasib (AMG-510) and adagrasib (MRTX849) were the earliest of which to enter the clinic [55,65,66]. AMG 510 is a first in class oral KRAS G12C inhibitor with evidence of clinical activity. Pre-clinical data of AMG510 demonstrated selective targeting of KRAS G12C tumors as monotherapy and in combination with cyto-toxic therapy [65]. MRTX849 is a potent and selective KRAS G12C inhibitor achieving 65% tumor regression in vitro and in patient-derived xenograft models [66,67].” (page 5, lines 201-218).

In addition, we modified the Figure 1 (page 3) and Figure 2 (page 9).

  1. In the liquid biopsy section, an extensive analysis of large NGS panels evaluated on liquid biopsy specimens should be carried out

RE: We would like to thank the reviewer for this comment. We have now added in the revised document, in Section 3, the following paragraph:

" Over the past few years, great technological advancements have been made and the highly sensitive and specific PCR-based techniques that were used so far for the detection of targeted genomic alterations are gradually replaced by high-throughput NGS techniques. NGS based methods in liquid biopsy offer a wider spectrum of molecular information obtained through a single analysis. Despite the higher cost, longer turn-around time and relatively lower sensitivity rates, NGS-based methods in liquid biopsy could positively affect the clinical management of NSCLC patients [85,86]. A multigene NGS approach in liquid biopsy is already included in the guidelines recently issued by the International Society for the Study of Lung Cancer (IASLC) and the European Society for Medical Oncology focused on personalized treatment of NSCLC patients, on providing biomarkers of prognostic significance during disease monitoring and on revealing the presence of alternative druggable alterations at progression of disease [87]. Although NGS-based assays are already performed in tissue samples, cfDNA analysis offers some advantages regarding the minimally invasive approach during disease progression and also depicts tumor heterogeneity. Very recently, FDA approved two NGS liquid biopsy tests, Guardant360 and FoundationOne Liquid CDx, based on the clinical utility of cfDNA testing for the personalized treatment of NSCLC patients. Many studies based on the NGS approach have been made and are presented in detail below." (page 6, lines 254-271)

  1. Section 3 appears too short. In my opinion, other relevant literature data should be implemented to clarify the integrating analytes found in torrent blood.

RE: We followed the reviewer suggestion and added more information about liquid biopsy in Section 3:
“3.1.1. CTCs

CTCs are cells that circulate in the bloodstream after separation from the primary tumor, potentially leading to metastasis formation under favorable conditions. The number of detected CTCs in 10 ml of blood ranges from 0 to over 10,000 [88] but detection of even single cells may suggest the presence of a developing neoplastic process [89–91]. Thus, CTC detection allows for the implementation of appropriate treatment faster, significantly increasing the chances of survival. In most cases, CTC analysis is based on the assessment of the number of cells and their phenotypic characteristics, with some of them enabling also their isolation and subsequent molecular testing [92–95]. However, all of them carry some challenges. One of the biggest problems may be the high heterogeneity of CTC, observed even in cells derived from the same tumor. The relatively low frequency of CTCs in relation to a large number of blood cells (usually a single CTC per 106-107 leukocytes) may be an additional difficulty [96]. For this reason, the proper analysis should be preceded by the enrichment stage, e.g., increasing the CTC concentration in the analyzed sample [97].

3.1.2. cfDNA

Another source of information about mutations and genetic changes within the tumor is cfDNA. cfDNA is present mainly in the bloodstream, as well as in other body fluids, such as urine, cerebrospinal fluid and pleural effusions. The analysis of peripheral blood-derived cfDNA can be performed in both serum and plasma. Studies have shown that the concentrations of cfDNA are higher in serum, however, it is not its preferred source due to the potential contamination with genomic DNA [98]. Although cfDNA is rapidly degraded by nucleases [99], it may reflect ongoing genetic changes in the tumor. It has been shown that the amount of cfDNA correlated with the tumor size [100], tumor stage [101] and the presence of metastases [102], suggesting that both primary and secondary genetic changes in the tumor are mirrored in cfDNA. This in-formation is particularly valuable in the aspect of monitoring the course and effectiveness of the applied therapy.

3.1.3. Exosomes

Exosomes are extracellular vesicles ranging in size from 30 to 100 nm. They are detected in the blood of patients with various types of cancer and consist, inter alia, of proteins and nucleic acids. By analyzing molecular changes in their elements (mutations, gene fusions or splicing variants), it is possible to determine, among others, tumor progression. The double lipid layer of exosomes makes it difficult to purify them, compared to CTCs and cfDNA, however, their structure provides better stability of their content, which allows better identification of the tumor's origin, genetic changes within it and potential resistance to treatment. Another advantage of exosomes is their much higher concentration in biological fluids, including blood, than in CTCs and cfDNA [103]. The analysis of isolated exosomes can be performed quickly and accurately with the use of electrochemical and fluorescent technologies, using binding with antibodies, aptamers or nanomaterials, with various test platforms [103].

3.1.4. TEP

Tumor-educated blood platelets are perhaps the newest ingredients in the liquid biopsy family. The concept of ‘platelet education’ by cancer refers to the presence of specific RNA signatures in platelets from patients with cancer and was first reported in 2010 [104]. Platelets are a fundamental component of the tumor microenvironment and are considered an important aspect of cancer biology as they contribute to tumor initiation, tumor progression, and therapy response. They create an environment supportive for neovascularization, reduce local tumor cell apoptosis and anoikis [105] and are able to induce the epithelial–mesenchymal switch in tumor cells, supporting metastatic spread. The process is generally based on providing mechanical and anti-NK cells protection (transfer of MHC I proteins) of CTCs by cell-fibrin-platelet aggregates [106].” (pages 6-7, lines 272-324).

Sincerely,

Joanna Budna-Tukan

Reviewer 2 Report

This is a comprehensive and well researched review that looks at the use of 'liquid biopsies' to monitor specific genomic aberrations within the context of NSCLC - an important area of ongoing development in clinical practice. 

Generally this is a good review, but I do have a few comments about the work that I think would improve it somewhat. 

Broadly, I think the piece is a little dense and overly wordy in places. There is some re-iteration and sentences are often so long they can be confusing. 

For example,

"Among the most common EGFR dependent resistance mechanisms that were detected in plasma, is EGFR C797S mutation that confers resistance to osimertinib [97,98]" 

Might read better re-phrased:

The EGFR C797S mutation, that confers resistance to osimertinib, is the most common resistance mechanism detected in plasma. 

This is only slightly shorter but easier to read (in my opinion). 

A re-write of the piece chopping out a lot of long sentences and repetition could probably cut the overall piece down significantly without losing any of the impact. 

You also need to be careful with some minor English/grammar:

"et al" should really be in italics

In your conclusion you state "Liquid biopsy is an increasingly popular diagnostic tool for the treatment of TKI-based NSCLC"

Except of course it isn't diagnostic but predictive. 

and

"The first clinical trials that compared the efficacy of 1st generation EGFR TKIs with chemotherapy generated the first promising results concerning predictive value of cfDNA 242 [84,85]."

I know what you are trying to say, but this long and wordy sentence is confusing, and appears to read that these were the first trials that compared EGFR TKIs to XCT, and that they happened to produce interesting data on cfDNA. Obviously the first trials pre-date these by some years!

You should be clear these are the first trials that utilise cfDNA whilst comparing EGFR TKI to XCT. 

Secondly, and perhaps more importantly, whilst this is a comprehensive review covering many studies, I never really know what the author's opinions are. I would suggest you are broadly in favour of liquid biopsies, but why? 

You cite other studies comparing them to tissue, but what are the best reasons? What do you like or not like? What are the major barriers to seeing it having more success? 

You state 

"the clinical validity of liquid biopsy has been proved, especially in lung cancer patients."

This is not quite true - liquid biopsy has potential in some areas, and is undeniably useful in some aspects, but it has not been proven to everyone. 

Indeed in the same paragraph you mention studies with mixed outcomes in this area.

Later you state:

"Interestingly, patients with no detectable EGFR mutations in plasma presented prolonged PFS due to lower tumor burden [95]."

In that case why bother testing at all? (I know why - but you should comment on this to the reader!) 

You briefly mention some of the different techniques of measuring cfDNA or CTCs. There are different assays and these have flaws and pros - but again your opinion and some commentary on this might be nice. 

You state they there has been some success - but what are pros and cons of each? 

How do they compare to tissue biopsy for e.g. time, price, accuracy etc etc?

Also if you can detect EGFR mutants by different means (exosomes, CTCs etc) do they all mean the same thing?

The evidence is not there yet to comment on this concretely - but what is your opinion? You should at least mention that this is a potential issue that needs to be addressed. 

I would suggest within the wider community 'liquid biopsies' are generally best used as a tool for monitoring patients with a known diagnosis for the reasons you mention.

It is a poor tool for screening or primary diagnosis compared to tissue at this point in time, and I think you should be clear on this. 

You mention NGS for specific aberrations e.g ALK and ROS1 with some good concordance - but can the whole profile for NSCLC be done? What are the limitations? How might you wish to see it compared to the current gold standard?

Overall this is a well researched and comprehensive piece - but I feel it could be significantly improved with a cut-down of the length by removing some of the longer sentences and repetition. This will give you space to give your opinion - which I (and the readers) want to hear! 

Author Response

Response to Reviewer 2

We would like to thank the Reviewer for the insightful and very constructive comments and suggestions that have guided us to significantly improve the quality of our review. We have revised the manuscript according to the comments and you will find below a point-by-point description of our modifications.

  1. Broadly, I think the piece is a little dense and overly wordy in places. There is some re-iteration and sentences are often so long they can be confusing.

For example,

"Among the most common EGFR dependent resistance mechanisms that were detected in plasma, is EGFR C797S mutation that confers resistance to osimertinib [97,98]"

Might read better re-phrased:

“The EGFR C797S mutation, that confers resistance to osimertinib, is the most common resistance mechanism detected in plasma. “

RE: Corrected according to Reviewer’s suggestion (page 10, lines 432-433).

According to Reviewer’s suggestion we also rephrased few sentences:

The phrase “It is recommended that, if possible, patients with stage I or II should be treated with complete surgical resection [1].” has been changed to: “Patients with stage I or II should be treated with complete surgical resection, if it is possible [1].” (page 2, lines 63-64).

The phrase “The treatment of lung cancer has recently evolved with the introduction of targeted therapy based on the use of tyrosine kinase inhibitors (TKI) in patients with metastatic NSCLC (stage III or IV) and certain gene alterations, including EGFR, ALK, ROS1, NTRK1, BRAF, HER2, and MET genes, which are, independent of PD-L1 levels.” has been changed to: “The treatment of lung cancer has recently evolved with the introduction of targeted therapy based on the use of tyrosine kinase inhibitors (TKIs). TKIs are used in patients with metastatic NSCLC (stage III or IV) and certain gene alterations, including EGFR, ALK, ROS1, NTRK1, BRAF, HER2, MET and KRAS genes, which are independent of PD-L1 levels.” (page 2, lines 73-77).

The phrase “Belonging to the class of non-selective type 1a inhibitors, crizotinib is a multi-tyrosine kinase inhibitor approved for the treatment of advanced NSCLC with ROS1 or ALK rearrangement.” has been changed to: “Crizotinib, non-selective type 1a inhibitor, is a multi-tyrosine kinase inhibitor approved for the treatment of advanced NSCLC with ROS1 or ALK rearrangement.” (page 5, lines 196-197).

The phrase “The first clinical trials that compared the efficacy of 1st generation EGFR TKIs with chemotherapy generated the first promising results concerning predictive value of cfDNA” has been changed to: “The first promising results on the predictive value of cfDNA in NSCLC patients were generated through important clinical trials that compared the efficacy of 1st generation EGFR TKIs against chemotherapy and included cfDNA analysis, beyond the classic approach of tissue biopsy [143,144].” (page 9, lines 398-401).

The phrase “Several studies have shown the concordance between plasma and tissue testing for the detection of T790M and suggested that noninvasive monitoring of resistance mutations could be useful for the detection of T790M missed because of tumor heterogeneity.” has been changed to: “Several studies have shown the concordance between plasma and tissue testing for the detection of T790M. Moreover, noninvasive monitoring captured tumor heterogeneity and identified resistance mutations that otherwise would have been missed while using classical tumor biopsy [150–152].” (pages 9-10, lines 416-418).

The phrase “Interestingly, patients with no detectable EGFR mutations in plasma presented prolonged PFS due to lower tumor burden [154].” has been changed to: “Interestingly, in this study, it was shown that in some patients that were positive for EGFR mutations in the primary tissue, a lack of EGFR mutations in plasma was observed and this was associated with better PFS; this could be possibly explained as a result of lower tumor burden that was shed in plasma [154].” (page 10, lines 425-428).

The phrase “Remarkably, greatest activity of the pertuzumab-erlotinib combination, with concomitant CTC changes, was seen in patients harboring EGFR mutations, and also high baseline CTC counts were associated with radiographic response, whereas high CTC counts are usually a poor prognostic factor [167].” has been changed to: “Remarkably, greatest activity of the pertuzumab-erlotinib combination, with concomitant CTC changes, was seen in patients harboring EGFR mutations., Despite that high CTC counts are usually a poor prognostic factor, in this study, high baseline CTC counts were associated with radiographic response [167]. (pages 10-11, lines 470-473).

The phrase “NGS plasma genotyping failed to detect ROS1 whereas it was found in tissue and corresponding CTCs by FISH analysis an important finding based on whether the patient benefited from crizotinib and ceritinib treatment in the long term [227].” has been changed to: “The patient benefited from crizotinib and ceritinib treatment in the long term based on the detection of ROS1 rearrangements in CTCs by FISH analysis whereas NGS plasma genotyping was negative [227].” (page 15, lines 697-699).

  1. You also need to be careful with some minor English/grammar: "et al" should really be in italics

RE: Corrected according to Reviewer’s suggestion.

  1. In your conclusion you state "Liquid biopsy is an increasingly popular diagnostic tool for the treatment of TKI-based NSCLC"

Except of course it isn't diagnostic but predictive.

RE: Corrected according to Reviewer’s suggestion (page 17, line 814).

  1. "The first clinical trials that compared the efficacy of 1st generation EGFR TKIs with chemotherapy generated the first promising results concerning predictive value of cfDNA 242 [84,85]."

I know what you are trying to say, but this long and wordy sentence is confusing, and appears to read that these were the first trials that compared EGFR TKIs to XCT, and that they happened to produce interesting data on cfDNA. Obviously the first trials pre-date these by some years!

You should be clear these are the first trials that utilize cfDNA whilst comparing EGFR TKI to XCT.

RE: We would like to thank the reviewer for this comment. We have now rephrased this sentence as follows and have now added it in the revised document, on page 9 lines 398-401 the following: "The first promising results on the predictive value of cfDNA in NSCLC patients were generated through important clinical trials that compared the efficacy of 1st generation EGFR TKIs against chemotherapy and included cfDNA analysis, beyond the classic approach of tissue biopsy [143,144]."

  1. Secondly, and perhaps more importantly, whilst this is a comprehensive review covering many studies, I never really know what the author's opinions are. I would suggest you are broadly in favor of liquid biopsies, but why?

RE: We followed the reviewer suggestion and added our opinion in Conclusions. Changes are seen on page 17 lines 813-815 (“Based on the presented research, we believe that liquid biopsy is an increasingly popular predictive tool in the treatment of NSCLC based on TKI.”) and lines 827-829 (“Despite such promising results obtained by many research teams we think that it is still necessary to carry out prospective studies on a larger group of patients to validate these methods before their application in clinical practice.”)

  1. You cite other studies comparing them to tissue, but what are the best reasons? What do you like or not like? What are the major barriers to seeing it having more success?

RE: We followed the reviewer suggestion and added our opinion in Conclusions. Changes are seen on page 17 lines 823-827 (“It is worth mentioning that tissue biopsy may be detrimental to the patient's health due to its invasiveness. Also, the quality of the available tumor biopsy and/or cytology material is not always adequate to perform the necessary molecular testing. Therefore, liquid biopsy can be a competitive predictive tool for tissue biopsy.”)

  1. You state

"the clinical validity of liquid biopsy has been proved, especially in lung cancer patients."

This is not quite true - liquid biopsy has potential in some areas, and is undeniably useful in some aspects, but it has not been proven to everyone.

Indeed, in the same paragraph you mention studies with mixed outcomes in this area.

RE: The phrase has been changed to: “However, the clinical validity of liquid biopsy has been proved, especially e.g., in lung cancer patients.” (page 7, line 329-330).

  1. Later you state:

"Interestingly, patients with no detectable EGFR mutations in plasma presented prolonged PFS due to lower tumor burden [95]."

In that case why bother testing at all? (I know why - but you should comment on this to the reader!)

RE: We would like to thank you for this comment. We have rephrased this sentence as follows: "Interestingly, in this study, it was shown that in some patients that were positive for EGFR mutations in the primary tissue, a lack of EGFR mutations in plasma was observed and this was associated with better PFS; this could be possibly explained as a result of lower tumor burden that was shed in plasma [154]." (page 10, lines 425-428)

  1. You briefly mention some of the different techniques of measuring cfDNA or CTCs. There are different assays, and these have flaws and pros - but again your opinion and some commentary on this might be nice

You state they there has been some success - but what are pros and cons of each?

How do they compare to tissue biopsy for e.g., time, price, accuracy etc.?

RE: This is true but this was not the main aim of our review, as this is also clear through the title. There are excellent reviews on the technologies used for CTC and ctDNA analysis. Since we already give them as references we prefer not to include a detailed description on the flaws and pros of each different technology.

  1. Also if you can detect EGFR mutants by different means (exosomes, CTCs etc.) do they all mean the same thing? The evidence is not there yet to comment on this concretely - but what is your opinion? You should at least mention that this is a potential issue that needs to be addressed.

 RE: We would like to thank you for this comment. In the revised manuscript we have now added in Section 4.1 the following:

Page 8 lines 372-378: "Most of studies performed so far have focused on ctDNA analysis for EGFR mutations. Analysis in CTCs is limited to a few studies, and exosomes even less. Moreover, there are only very few studies that have compared EGFR mutations in CTCs and ctDNA directly in the same clinical samples using the same blood draws and the same methodologies. Based on this we believe that the analysis of EGFR mutations in exosomes provides clinically important information that needs to be confirmed through larger clinical studies."

Page 11 lines 497-500: “During the last years, the analysis of circulating exosomes has provided new opportunities for cancer diagnosis and monitoring of disease progression in the liquid biopsy field. Many studies in NSCLC have shown higher sensitivity and specificity while combining exosomes with cfDNA testing, thus implicating their clinical utility" [176].”

Pages 11-12 lines 521-524: “Exosomes carry important molecular information since they are secreted from living cells and in some cases their analysis could be more informative than cfDNA [182]. However, there are still challenges to overcome and further research need to be done in order to integrate exosome analysis in the liquid biopsy setting”

  1. I would suggest within the wider community 'liquid biopsies' are generally best used as a tool for monitoring patients with a known diagnosis for the reasons you mention.

It is a poor tool for screening or primary diagnosis compared to tissue at this point in time, and I think you should be clear on this.

RE: We would like to thank you for this comment. We have now added the following in Section 3:

“Despite some significant efforts that have been made so far to develop methods for early cancer detection by using ctDNA, presenting promising results, such as CancerSeek [78] and very recently the Galleri Test (Grail) [79], liquid biopsy is not established as a tool yet for early diagnosis.” (Page 6 lines 246-249)

  1. You mention NGS for specific aberrations e.g., ALK and ROS1 with some good concordance - but can the whole profile for NSCLC be done? What are the limitations? How might you wish to see it compared to the current gold standard?

RE: We would like to thank the reviewer for this comment. We have now added in the revised document, in Section 3, the following paragraph:

“Over the past few years, great technological advancements have been made and the highly sensitive and specific PCR-based techniques that were used so far for the detection of targeted genomic alterations are gradually replaced by high-throughput NGS techniques. NGS based methods in liquid biopsy offer a wider spectrum of molecular information obtained through a single analysis. Despite the higher cost, longer turn-around time and relatively lower sensitivity rates, NGS-based methods in liquid biopsy could positively affect the clinical management of NSCLC patients [85,86]. A multigene NGS approach in liquid biopsy is already included in the guidelines recently issued by the International Society for the Study of Lung Cancer (IASLC) and the European Society for Medical Oncology focused on personalized treatment of NSCLC patients, on providing biomarkers of prognostic significance during disease monitoring and on revealing the presence of alternative druggable alterations at progression of disease [87]. Although NGS-based assays are already performed in tissue samples, cfDNA analysis offers some advantages regarding the minimally invasive approach during disease progression and also depicts tumor heterogeneity. Very recently, FDA approved two NGS liquid biopsy tests, Guardant360 and FoundationOne Liquid CDx, based on the clinical utility of cfDNA testing for the personalized treatment of NSCLC patients. Many studies based on the NGS approach have been made and are presented in detail below.” (Page 6, lines 254-271)

Sincerely,

Joanna Budna-Tukan

Round 2

Reviewer 1 Report

The manuscript may be accepted in the present form